# Evidence of Chronic Complement Activation in Asymptomatic Pediatric Brain Injury Patients: A Pilot Study

**DOI:** 10.3390/children10010045

**Published:** 2022-12-26

**Authors:** Scott A. Holmes, Joud Mar’i, Jordan Lemme, Anne Margarette Maallo, Alyssa Lebel, Laura Simons, Michael J. O’Brien, David Zurakowski, Rami Burnstein, David Borsook

**Affiliations:** 1Department of Anesthesia, Critical Care and Pain Medicine, Pediatric Pain Pathway Lab 1, Boston, MA 02215, USA; 2Pain and Affective Neuroscience Center, Boston Children’s Hospital, Boston, MA 02215, USA; 3Department of Anesthesia, Critical Care and Pain Medicine, Boston Children’s Hospital, Boston, MA 02215, USA; 4Department of Anesthesia, Stanford University, Stanford, CA 94305, USA; 5Sports Medicine Division, Boston Children’s Hospital, Boston, MA 02215, USA; 6Department of Anesthesia, Beth Israel Deaconness Medical Center, Boston, MA 02215, USA; 7Department of Radiology, Massachuetts General Hospital, Boston, MA 02114, USA

**Keywords:** inflammation, brain injury, pediatrics

## Abstract

Physical insult from a mild Traumatic Brain Injury (mTBI) leads to changes in blood flow in the brain and measurable changes in white matter, suggesting a physiological basis for chronic symptom presentation. Post-traumatic headache (PTH) is frequently reported by persons after an mTBI that may persist beyond the acute period (>3 months). It remains unclear whether ongoing inflammation may contribute to the clinical trajectory of PTH. We recruited a cohort of pediatric subjects with PTH who had an acute or a persistent clinical trajectory, each around the 3-month post-injury time point, as well as a group of age and sex-matched healthy controls. We collected salivary markers of mRNA expression as well as brain imaging and psychological testing. The persistent PTH group showed the highest levels of psychological burden and pain symptom reporting. Our data suggest that the acute and persistent PTH cohort had elevated levels of complement factors relative to healthy controls. The greatest change in mRNA expression was found in the acute-PTH cohort wherein the complement cascade and markers of vascular health showed a prominent role for C1Q in PTH pathophysiology. These findings (1) underscore a prolonged engagement of what is normally a healthy response and (2) show that a persistent PTH symptom trajectory may parallel a poorly regulated inflammatory response.

## 1. Introduction

The neuroinflammatory response to brain injury occurs secondary to the primary physical disruption of axons and underlies efforts at remodeling and a return to homeostasis. Traumatic brain injury is a leading cause of morbidity and mortality in the world [1] and occurs along a spectrum of mild to severe. Post-traumatic headache (PTH) is one of the most prominent and debilitating side effects of brain injury [2] and may persist years after injury. Although significant pathology and network alterations have been noted across the cerebrum and in sub-cortical structures, there is notable involvement of the thalamus in headache symptom pathophysiology and headache-related pain [3,4,5,6,7]. It is unclear how some individuals can achieve asymptomatic status while others have ongoing symptoms. Persistent inflammation, known to be present in other headache conditions [8], maybe one such contributor to persistent PTH. 

The complement cascade is an inflammatory pathway associated with assisting in the identification of foreign tissues and the removal of debris through angiogenic and inflammatory processes. Multiple complement-related molecules are noted to be upregulated in cases of severe TBI, including C3/CFB/C1Q, C4, C9, C5b9, and MBL [9,10,11,12]. Review work on moderate and severe traumatic brain injury (TBI) supports a prominent role for the complement cascade system after injury in terms of symptom burden [13]. Complement activation, which can be achieved by binding of C1q to apoptotic cells [14,15], is associated with smooth wound healing [16] in mouse models and proper regulation of homeostasis, including removal of cell debris from injury [17]. Using mouse models of brain injury, inhibition of C1, an initiator of the complement system, has been shown to reduce motor deficits, cognitive dysfunction, and contusion volumes [18]. Inhibition of C3, which has relationships with multiple complement pathways, can reduce the extent of neuronal death and chronic inflammation and improve cognitive recovery after brain injury [19,20]. Similarly, inhibition of confirmatory factor B (CFB) is associated with attenuated cell death [21]. As such, multiple components of the complement pathway have been implicated in brain injury using mouse models. No research to date has evaluated complement activity in pediatric mTBI.

In this hypothesis-generating investigation, we evaluated saliva-based mRNA expression from 48 participants equally divided between healthy controls and persons with acute and persistent PTH at three months post-injury. Post-traumatic headache is a chronic pain condition experienced by a high percentage of pediatric patients after brain injury [22] and may be triggered through inflammatory pathways (Mayer et al., 2013), implicating vascular and neurological domains. Although in some patients, headache symptoms resolve within a week or two after brain injury, in others, symptoms persist and may extend years after injury. In this investigation, we aimed to: (1) evaluate changes in transcriptional expression of mRNA involved in the complement cascade between PTH cohorts and healthy controls at three months post-injury, (2) explore the most dominant contributors towards observed group differences, and (3) explore the relationship between complement activation and cerebral blood flow from the thalamus. We performed whole-brain arterial spin labeling and diffusion tensor imaging to outline the evidence for vascular change and white matter pathology. We hypothesized that persons with persistent symptoms would display a unique inflammatory profile relative to persons with acute symptoms that would inform patient trajectories. 

## 2. Methods

Participants were recruited from the greater Boston area as part of a 5-year NIH-funded study on Post-Traumatic Headaches in pediatric cohorts. Clinical participants were recruited from the greater Boston area or from PowerChart review of patient records for patients visiting the Sports Medicine Clinic at Boston Children’s Hospital (Waltham, MA, USA). Healthy controls were recruited via community advertisements, including flyers that were hung around college campuses in the Longwood Medical Area and on the Partners HealthCare Portal. All patients with post-traumatic headaches were screened by a study physician and were found to fulfill the International Classification of Diseases, Ninth Revision, and reported to have developed a headache within seven days after their mTBI. The acute cohort were participants who had developed PTH and had their self-reported symptoms resolved within 2 weeks to 1 month after injury (A-PTH). Participants with PTH who had self-reported symptoms persisting beyond 1-month were placed in the persistent group (P-PTH). Participants were screened for the use of recreational and illicit drug use prior to study enrollment.

A total of 32 participants with PTH completed this investigation and were equally divided between the A-PTH and P-PTH cohorts. A group of 16 age and sex-matched healthy controls were included. Four subjects had to be removed because of sample quality (mRNA did not reach the level necessary for quantification), leaving 15 participants with A-PTH, 15 with P-PTH, and 14 with Healthy Controls. All subjects were right-handed and between the ages of 12–24 at the time of the study visit and had no significant history of pre-existing headaches, chronic pain, psychiatric or neurological conditions (including clinical depression or anxiety). Enrollment includes screening for drugs of abuse, including barbiturates, benzodiazepines, amphetamines, tetrahydrocannabinol, and medications that would interfere with study findings. This study was approved by the Institutional Review Board at Boston Children’s Hospital and conducted in accordance with the principles of the Declaration of Helsinki. Informed consent and assent were obtained from all subjects prior to enrollment. 

### 2.1. Psychological Questionnaires and Testing

At the 3-month study visit (3 months from the date of mTBI +/- 30 Days), the participant met with a physician or registered nurse to review eligibility and mTBI symptomology. The following questionnaires were completed by both PTH and control cohorts: Depression (Childhood Depression Inventory [23] (CDI: age < 18; Beck Depression Inventory (BDI: age > 18; [24]), revised children’s manifest anxiety scale (RCMAS [25];), pubertal developmental scale (PDS; [26], catastrophizing pain scale (PCS; [27]), fear of pain questionnaire (FOPQ; [28]), and pediatric pain screening tool (PPST; [29]). T-scores were extracted and used from the CDI and BDI and integrated into the Depression metric. The PTH cohort received two additional questionnaires: Rivermead Concussion Survey [30] and to assess changes in sensitivity to painful stimuli. 

### 2.2. Magnetic Resonance Imaging Data Acquisition

Magnetic resonance imaging (MRI) data were collected on a 3T Siemens Magnetom TrioTim scanner with a 12-channel phased array head coil (Erlangen, Germany). T1 magnetization-Prepared Rapid Acquisition Gradient-Echo (MPRAGE) anatomical images were collected using a gradient echo-planar pulse sequence with 1.0 × 1.0 × 1.0 mm resolution. MPRAGE scan parameters consisted of the following: repetition time (TR) = 2520 msec; echo time (TE) = 1.74 3.54, 5.34, 7.14 msec; field of view (FOV) = 220 × 220; flip angle (FA) = 7°; and axial slices = 176. RS-Fc data were collected using a gradient echo-echo planar pulse sequence with 3.0 mm × 3.0 mm × 3.0 mm resolution. Functional MRI (fMRI) scan parameters consisted of the following: TR = 1100 msec; TE = 30 ms; FOV = 228 × 228; FA = 70°; axial slices = 51; volumes = 320; and acquisition time = 6 min. Patients were instructed to remain still, clear their minds and keep their eyes open during the scan sequence.

### 2.3. ASL Data Processing

We performed pseudo-continuous arterial spin labeling (pCASL) with the following parameters: multi-slice single-shot GE-EPI readout, PLD = 1.3 s, labeling duration = 1.5 s, TR/TE = 3870/12, FOV = 220 mm^2^, matrix = 64.64, 26 slices, slice thickness = 5 mm). Two calibration images (no labeling or background suppression, all other parameters identical to the pCASL scan) were collected to enable the estimation of the equilibrium magnetization of blood. The BASIL toolbox from FSL was used to process pCASL data. The “asl_file” tool was used for ASL data manipulation according to the number of inflow times (nits) = 1.2; iaf = tc—the input data were label-control (tag-control) pairs; diff—a pairwise subtraction of odd volumes from even volumes. There was N/2 in the output images for N volumes in the input images, and we took the mean over the volumes for the creation of the mean difference image. The “oxford_asl” automated utility was used to process ASL data to produce a calibrated map of resting state tissue perfusion: tr = 3.89, cgain = 1, t1csf = 4.3, t2csf = 750, t2bl = 150, te = 0, bat = 1.3, t1 = 1.3, t1b = 1.65, alpha = 0.85, fixbolus, motion correction, partial volume correction, and regional analysis. 

### 2.4. Diffusion MRI

One set of multi-shell HARDI images was acquired (2mm isotropic resolution, TE = 10 ms, TR = 5200 ms, TA = 7:37, 30 directions each at b = 1000 s/mm^2^ and b = 2000 s/mm^2^; 4 b = 0 images). All image processing was performed through the Boston Children’s Hospital’s High-Performance Computing Resources Cluster Enkefalos. Software used in the project was installed and configured by BioGrids [31]. Statistical analyses were performed using Matlab 2019b (www.mathworks.com (accessed on 19 December 2022)). All HARDI images were preprocessed using Mrtrix v3.0.1, 64-bit release version, built 1 July 2020, using Eigen 3.3.7 [32] with the following steps: removal of noise [33,34,35], Gibbs ringing artifact [36], susceptibility-induced distortions [37,38,39], and B1 field inhomogeneity [32]. The HARDI images were upsampled to 1.25 mm isotropic resolution. The upsampled images were used to estimate the tensor and hence, the various microstructural indices: fractional anisotropy (FA), axial diffusivity (AD), mean diffusivity (MD), and radial diffusivity (RD). Likewise, the upsampled images were processed through TractSeg [40] for the automated delineation of the white matter tractograms. The underlying indices were sampled for each tractogram, generating four white matter microstructure values used in the analyses.

### 2.5. Transcriptome Profiling on Saliva Samples

Transcriptome profiling was evaluated on saliva samples provided at the time of the study visit. Samples were obtained through Oragene-provided saliva collection tubes (Oragene, Ottawa, CA, USA) and shaken prior to storage at −80deg F while they awaited processing. Removing from storage included heating for one hour at 50 degrees and included B-mercaptoethanol to buffer the sample (2 mL). RNA was extracted using a total RNA preparation kit from Qiagen Biotechnology Company (Hilden, Germany). The quality and quantity of mRNA were performed using Agilent (Santa Clara, CA, USA) Bioanalyzer. Only high-quality mRNA (Distribution Value of 200 or higher was used for transcriptome profiling).

### 2.6. Nanostring Gene Expression Analysis

Samples were transferred to Nanostring services for bioinformatics and processed on a nCounter digital analyzer (NanoString^TM^, Seattle, WA, USA). The raw data were imported and evaluated in nSolver4.0 (NanoString^TM^) to ensure data quality and perform normalization. Statistical analysis was performed by the Advanced Analysis package2.0 (NanoString^TM^). Only samples listed as members of the initiating and regulation component of the complement cascade were used for this analysis. Statistical analyses were performed on log2 transformed counts. Individual participant mRNA expression was found to meet expression thresholds if it exceeded the level of log2 expression = 1.

### 2.7. Statistical Analysis

Behavioral data were evaluated using independent sample t-tests comparing persons with A-PTH, or P-PTH relative to healthy controls. We elected to perform independent sample t-tests in lieu of omnibus tests based on the sample size and our primary interest in understanding clinical differences relative to our healthy control cohort. Whole brain analyses were performed for cerebral perfusion and white matter integrity with a statistical threshold of 0.05. Exploratory analyses were conducted using group-level data and Pearson correlation analyses to evaluate trends. To explore candidate markers for future exploration, we performed Principal Component Analysis (PCA) on the expressed mRNA markers from the complement cascade and evaluated component output using scree and cumulative variance plots. We extracted the top five markers of the complement cascade from the first component (representing the greatest amount of variance in the dataset relative to other components) and performed an independent sample t-test between our clinical cohorts and healthy controls. A *p*-value < 0.05 was deemed statistically significant. Based on the sample size and exploratory nature of the study, we did not correct for multiple comparisons.

## 3. Results

### 3.1. Participant Demographics

Due to sample processing standards, four participants (HC: 2; Acute: 1; Persistent: 1) had to be removed from the analysis because of insufficient sample yield for quantification. No group difference were observed for age, F(2) = 0.894, *p* = 0.416, or sex, X2 = 1.2, *p* = 0.54, between the three cohorts. There were no group differences found between healthy controls and persons with resolved PTH symptoms (A-PTH) regarding psychological or symptom-related questionnaires (ps > 0.05). The persistent PTH cohort had higher values than healthy controls on the FOPQ-C, t(29) = 3.004, *p* = 0.005, and PPST, t(29) = 3.142, *p* = 0.004. The P-PTH cohort was found to report greater overall mTBI symptoms, t(30) = 7.0, *p* < 0.001, and headache symptoms, t(30) = 4.326, *p* < 0.001, relative to the resolved PTH cohort. Comparisons were not performed between the A-PTH and P-PTH cohorts in line with the objectives of this investigation and our limited sample sizes. Findings are presented in Figure 1 and Table 1.

### 3.2. Neuroimaging

An analysis of arterial spin labeling whole brain data showed that cerebral blood flow (CBF) was greater in the A-PTH relative to the healthy control cohort in the left, t(26) = 2.75, *p* = 0.01, and right, t(26) = 2.14, *p* = 0.042, lateral ventricles. Neither the left nor right lateral ventricles were found to differ between persons with persistent PTH and healthy controls. Whole brain analysis of white matter fibers showed that the A-PTH cohort had decreased axial diffusivity in the right fornix (*p* = 0.0018) and the left striato-fronto-orbital (*p* = 0.0495), as well as decreased mean diffusivity of the right fornix (*p* = 0.0065), relative to the healthy control cohort. For the P-PTH cohort, the right fornix showed decreased AD (*p* < 0.0001) and MD (*p* = 0.004), and RD (*p* = 0.03) relative to healthy controls.

### 3.3. Complement Cascade: Pathway Analysis

In the A-PTH cohort, members of the initiation component of the complement cascade that was above significant levels of expression relative to healthy controls were C4A, C1S, CFB, C1QB, C3, C1QA, CRP, whereas only C1QA was found to be above significance in the P-PTH cohort (Figure 2). Analysis of complement regulatory molecules demonstrated upregulation of CD55, C4A, C1S, and CFB in the A-PTH cohort, whereas four mRNA were expressed in the P-PTH cohort: CD55, C5, C1QA, and C8B. Notably, there is overlap as some mRNA are involved in the initiation and regulation component of the complement cascade.

### 3.4. Complemented Mediated Activity Driven by Core Set of mRNA

The top five candidate markers from the first component (Greater than the loading of 3) were retrieved (Figure 3 for PCA findings and Figure 4 for group outcomes). This gave five target molecules that were evaluated between PTH cohorts and healthy controls. Relative to healthy controls, the A-PTH cohort had greater expression of C1QA, t(26) = 2.43, *p* = 0.022, C1QB, t(26) = 2.16, *p* = 0.04, C4A, t(25.6) = 2.76, *p* = 0.01, C8B, t(25.6) = 2.15, *p* = 0.04, and CFB, t(24.6) = 3.077, *p* = 0.005. For the P-PTH cohort, only C1QB, t(26) = 2.16, *p* = 0.04, and C8B, t(26) = 2.15, *p* = 0.04, were found to be significantly more expressed than healthy controls (Figure 4).

We elected to explore relationships between cerebral blood flow, symptom presentation, and inflammation in relation to the left and right thalamus based on the thalamus displaying hyperconnectivity in early stages after mTBI [41] and being a target for C1Q-mediated inflammation [42] after cortically-focused brain injury (mouse model). As shown in Figure 5, there is a trend towards decreased CBF in the A-PTH cohort and more variable CBF in the P-PTH cohort. For the P-PTH cohort, CBF had a positive relationship with self-reported headache symptoms. In relation to the top five inflammatory markers, there appeared to be a shift in the canonical relationship between mRNA expression and CBF that progressed in the healthy controls to A-PTH and was greater in the P-PTH cohort.

## 4. Discussion

Prior research has shown how both how overt behavioral symptoms may not accurately reflect underlying central nervous system processes [43,44] and how inflammation from an mTBI (in adults) may extend through the first year after injury [45]. The complement cascade plays a critical role in canonical neurological development, including homeostatic functions, synaptic pruning, and regulating inflammation [46]. There has been no research to date evaluating the role of complement in pediatric mTBI. We provide evidence to suggest (1) the presence of elevated complement activity in persons regardless of symptom reporting and (2) a dissociable pattern of complement activation and regulation between symptomatic and asymptomatic participants. We discuss findings in terms of a potential mechanism for persistent headaches.

### 4.1. Elevated Complement Activity in Pediatric Subjects with PTH

This is the first study to demonstrate complement upregulation in pediatric participants with a mild traumatic brain injury. A brain injury is a catalyst for a cascade of neuroinflammatory events aimed at removing damaged tissue and preserving neurological function [47]. Levels of C1Q (C1QA and C1QB) in both A-PTH and P-PTH cohorts are suggestive of a mounting immune response. This would align with data from rodent models, where complement mediates neuroinflammation and cognitive decline and shows continued activity at six months after brain injury [48]. The role of C1Q is to initiate an inflammatory response and can have both survival and apoptotic functions [49], with evidence of C1Q tagging weaker synapses for removal in stroke models [50]. Prior work has shown how sub-units of the C1Q complex, including C1QA, C1QB, and C1QC, are associated with modulating the local environment of tumor cells [51] and may therefore reflect local microenvironment remodeling after brain injury. To date, the only other study—to the knowledge of the authors—to evaluate the complement cascade in pediatrics shows a profound effect of C1Q, C3, and C4 in pediatric subjects with systemic lupus erythematosus, which, notably, is tied to autoimmune activity and headache symptoms [52]. This finding suggests a role for immune dysregulation in persistent PTH symptoms. Findings from the PCA analysis also showed a strong role of C4A, C8B, and CFB. C8B is a compound that forms a component of the membrane attack complex [53], and elevated levels have been associated with the development of psychiatric disorders in pediatric cases [54] and adult cases of schizophrenia [55] and major depression disorder/bipolar disorder [56]. Observations of CFB in CSF of brain-injured patients are suggestive of ongoing inflammatory processing from secondary brain injury [9]. Together, we show evidence of both initiation and sustained inflammation in the complement cascade pathway in both PTH cohorts, likely resulting from brain trauma.

### 4.2. Dissociating PTH Cohorts and the Role of Symptom Presentation

The primary distinction between PTH cohorts appears to be in downstream effectors of the complement cascade. Downstream expression of mRNA, including C3, C4A, and C8A, show greater levels in the A-PTH than P-PTH cohort when compared with healthy controls. Interestingly, expression of C3 may be associated with protection against NMDA neurotoxicity [57]; however, it has also been associated with degeneration and exacerbation of chronic deficits in long-term models of TBI 20. The inhibition of C3 is implicated in preventing chronic inflammation, perhaps suggesting that current observations in the P-PTH cohort reflect a suppressed inflammatory response connected with C3 inhibition [58]. Timing of events is also a notable concern after brain injury. Both C1 and C3 have been associated with pro-apoptotic environments [59,60] and may exhibit temporal sensitivity where C3a expression in the late subacute period may assist in recovery [61]. Expression of CD55 (seen in both cohorts) may inhibit C3 convertases and protect neovascular processes from complement-mediated removal [62]. A lack of C3 in the persistent cohort may therefore reflect aberrant vascularization that has not been effectively removed and may contribute towards symptom expression. This could align with observations of increased cerebral blood flow in the lateral ventricles in the A-PTH cohort and a more effective means of effectively clearing damaged cells in this cohort. Alternatively, ineffective clearance in persons with P-PTH may align with observations of a buildup of complement, which has concerns for neurodegeneration, as seen in the choroid plexus of the retina [63,64]. Findings align with published work on this PTH cohort, where we show more extensive functional network rearrangements using resting-state fMRI in persons with A-PTH, relative to P-PTH, suggesting that the expression of pro-vascular factors contribute towards adaptive network rearrangements [44]. Differences in the expression of complement may underscore the persistence of PTH symptoms.

### 4.3. The Role of Complement in Brain Injury and PTH Symptom Persistence

Raw observations and quantification of complement activity are difficult to achieve, derived mostly from post-mortem tissues and cerebral spinal fluid samples. In adult most-mortem tissues in persons who had sustained a TBI, there is evidence of elevated—relative to healthy controls—levels of C1Q, C3b, C3d, and Membrane Attack Complex (MAC) in the penumbral regions of injured tissue [65]. Similarly, elevated levels of C3 and C1q [9,11], Mannose Binding Lectin (MBL) [66], and MAC [10] have been observed in the CSF of TBI patients relative to healthy controls (see [47] for review). These align with the current findings in supporting the expression of C1Q after brain injury, in addition to other complement factors. We extend these findings by showing the presence of complement expression in pediatric mild traumatic brain injury and their possible role in symptom persistence.

Complement activation appears to play a role in PTH after pediatric mTBI. We draw attention to the vascular nature of sustained complement activity. Observing a negative correlation between C1Q and blood flow—and Thalamus—that is unique from healthy controls, and persons with acute symptoms imply C1Q may be associated with distinct processes in each mTBI cohort. C1Q is a strong driver of vascular changes, independent of its role in complement activation [67]. Notably, the complement has a particular role in regulating hypertension [68] which directly implicates the brain. In the context of headaches, it is notable that C3 and C5 are both associated with hypertension [68,69,70], suggesting that the headache from PTH may have hypertensive qualities when C3 and C5 are both elevated. This is as well supported by observations that C3a and C5a are both known anaphylatoxins and can destabilize vascular permeability in models, including pre-eclampsia [71]. Moreover, C4A has been shown to impact endothelial permeability through PAR1/4 [72]. Ongoing expression of C1Q above normal in both mTBI cohorts suggests persistent inflammatory activity in this pediatric cohort that is not found in healthy individuals. This is a significant finding, particularly in pediatrics, as sustained and above-normal complement activity may be associated with dysregulated vascularization and overly aggressive synaptic pruning during neurodevelopment [49,73]. As properly regulated complement activity is required for smooth wound healing [16], destabilization, as shown in the current cohorts, may contribute to persistent symptoms after pediatric brain injury.

The above work has limitations that require mention. (1) Our sample cohorts were defined through self-reported symptoms. This is a fundamental limitation to pain and headache research and may present limitations based on individual pain thresholds or their recall of headache events. (2) Our cohort sizes were relatively small for brain imaging research and were defined by inflammation. Despite the limited sample size for brain imaging conclusions, we are the first to evaluate both acute and persistent pediatric mTBI participants for in vivo inflammation. (3) We are evaluating transcriptional products. As such, although an expression of a certain mRNA can be quantified, it is unclear from this investigation if these molecules proceed fully and efficiently into their protein through translational mechanisms.

The current investigation demonstrates (1) persistent activation of the complement cascade in pediatric persons with PTH symptoms and (2) a potential biological basis for prolonged headache symptoms from an mTBI. These findings are interesting, considering that complement activity is usually seen in conditions of autoimmunity or infection with bacterial or viral agents. It will be important to consider translational efforts to integrate molecular markers with neuroimaging and behavioral research to improve patient care.

## Figures and Tables

**Figure 1 children-10-00045-f001:**
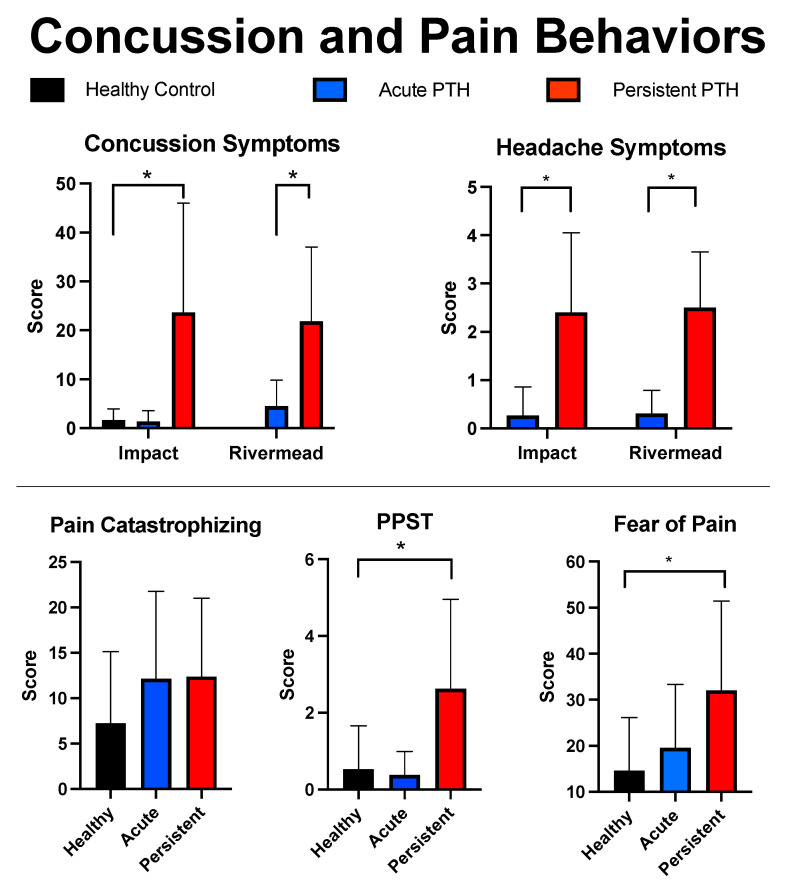
Self-reported symptoms. Findings are outlined for the three cohorts using mean and standard deviations for the main mTBI and pain questionnaires. mTBI symptom reporting is provided for the Impact and Rivermead mTBI surveys. * indicates a *p*-value less than 0.05.

**Figure 2 children-10-00045-f002:**
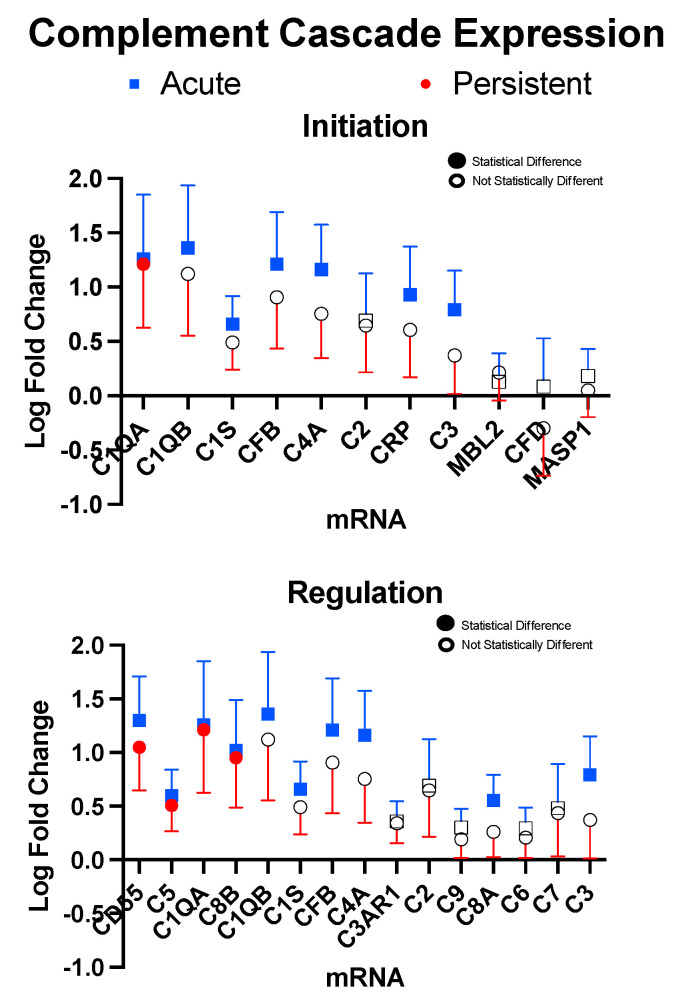
Initiation and Regulation of the complement cascade. Markers are presented for factors that were found for at least one cohort to be above detectable limits. Samples were found to be statistically significant (filled boxes or circles) if they had *p*-values < 0.05. Error bars represent one standard deviation.

**Figure 3 children-10-00045-f003:**
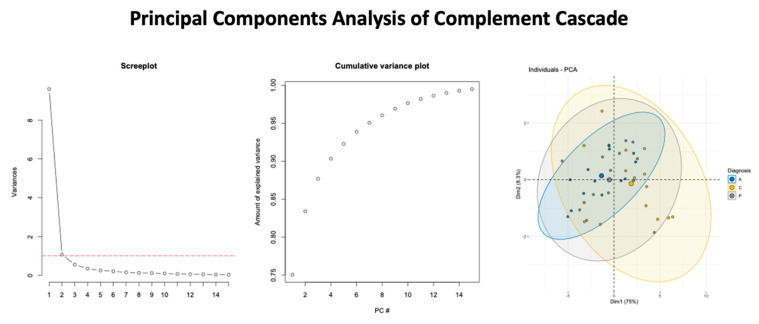
Summary of findings from the Principal Component Analysis. The screeplot (**left panel**), cumulative variance plot (**middle panel**), and the variance from the first two dimensions (**right panel**) with ellipses to identify the respective study cohorts are shown.

**Figure 4 children-10-00045-f004:**
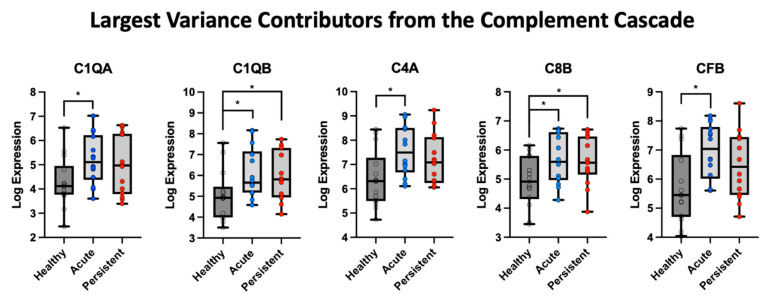
Five highest contributions to the first component from the PCA analysis of the complement cascade. The log expression values were evaluated in each cohort relative to healthy control data, with * indicating a *p*-value less than 0.05.

**Figure 5 children-10-00045-f005:**
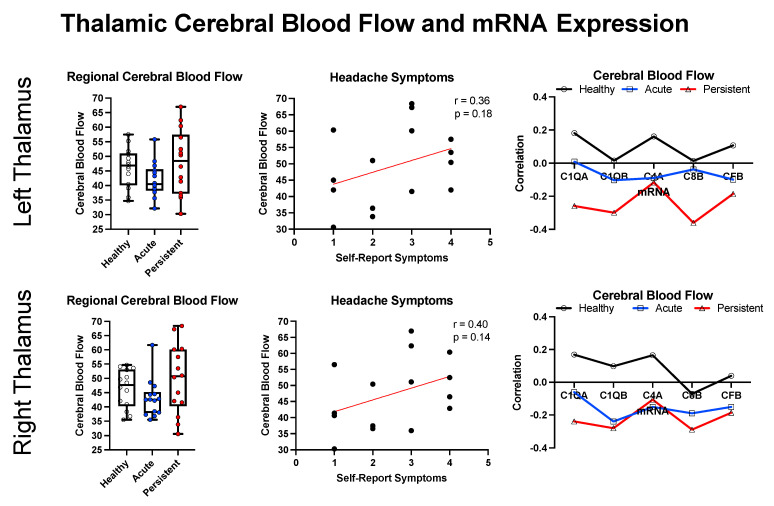
Regional cross-modal analysis for the bilateral thalamus. Data is presented for cerebral blood flow (**left**), the correlation between cerebral blood flow and self-reported headache symptoms (**middle**), and the correlation between CBF and inflammatory markers (**right**).

**Table 1 children-10-00045-t001:** Participant demographics. An overview of participant demographics is provided, outlining participant characteristics as well as mTBI-related characteristics.

	Healthy Control (*n* = 14)		Acute PTH (*n* = 15)		Persistent PTH (*n* = 15)	
	M	SD	M	SD	M	SD
Time Since Injury (days)	-	-	105.625	12.03	101	12.36
Age (years)	16.47	2.48	16.2	2.52	15.4	2.17
Females (#)	8	50	9	56	11	69
Rivermead Scale	-	-	4.5	5.3	21.9	15.2
Pain Catastrophizing Scale	7.26	7.87	12.125	9.64	12.375	8.64
Fear of Pain	14.6	11.52	19.56	13.8	32	19.45
Pediatric Pain Screening Tool	0.53	1.125	0.375	0.62	2.625	2.33
Impact Scale	-	-	1.4	2.16	23.7	22.4

## Data Availability

Data can be accessed by contacting SH and is available upon request.

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
