# Peer review of "Evidence of Chronic Complement Activation in Asymptomatic Pediatric Brain Injury Patients: A Pilot Study"

_children, 2022, doi:10.3390/children10010045_

Round 1

Reviewer 1 Report

-Multiple spelling and grammatical errors. You used R-TBI instead of A-TBI. Please correct/clarify. 

-Please provide abbreviations in brackets so the reader can refer to them for clarification. 

-Would like to see references related to the current evidence on specific complement markers in adults. Are there other studies that reflect, support, or refute your findings in adults? Please include in one paragraph under discussion

- The other pediatric study in the same arena was outlined, but it would help to organize the main differences and similarities in one paragraph instead of spreading it out. It would be good for the reader to understand how the current evidence relates to your study

-The whole article is well organized and easy to read except for the discussion. Splitting it up into sub-headings or even different paragraphs would be helpful for the reader. 

Author Response

Reviewer 1:

Multiple spelling and grammatical errors. You used R-TBI instead of A-TBI. Please correct/clarify. 

Thank you. We have had the manuscript reviewed and edited. You will note multiple track changes to the document where we have edited spelling mistakes or attempted to improve the grammar / structure of the document.

Please provide abbreviations in brackets so the reader can refer to them for clarification. 

We have edited the manuscript to align with reviewer comments.

Would like to see references related to the current evidence on specific complement markers in adults. Are there other studies that reflect, support, or refute your findings in adults? Please include in one paragraph under discussion

We have provided a new paragraph in the discussion (which has been re-structured) to improve clarity and align with reviewer comments. We provide an overview of what has been observed in the adult literature and how that aligns with current observations.

Raw observations and quantification of complement activity is difficult to achieve, derived mostly from post-mortem tissues and cerebral spinal fluid samples. In adult most-mortem tissues in persons who had sustained a TBI, there is evidence of elevated – relative to healthy controls - levels of C1Q, C3b, C3d, and Membrane Attack Complex (MAC) in the penumbral regions of injured tissue 65. Similarly, elevated levels of C3 and C1q 66,67, Mannose Binding Lectin (MBL) 68 and MAC 10 have been observed in the CSF of TBI patients relative to healthy controls (see 69 for review). These align with the current findings in supporting the expression of C1Q after brain injury, in addition to other complement factors. We extend these findings by showing the presence of complement expression in pediatric mild traumatic brain injury, and their possible role in symptom persistence.  

The other pediatric study in the same arena was outlined, but it would help to organize the main differences and similarities in one paragraph instead of spreading it out. It would be good for the reader to understand how the current evidence relates to your study

Thank you for your comment. We have restructured the discussion so there is now one paragraph about similarities, a second about differences, and then subsequent paragraphs provide more abstract information about how our findings fit within the larger literature.

The whole article is well organized and easy to read except for the discussion. Splitting it up into sub-headings or even different paragraphs would be helpful for the reader. 

Thank you. We have provided the sub-headings as suggested and feel this greatly improves the structure of the manuscript.

Reviewer 2 Report

The authors analyzed the evidence of chronic complement activation in 30 asymptomatic pediatric brain injuries. 

It is a well-planned study and I think it will contribute to the literature.

However, there are concerns particularly for technical and statistical issues.

1. The authors stated "Four subjects had to be removed because of sample qual-88 ity, leaving 15 participants with A-PTH, 15 with P-PTH, and 14 Healthy Controls." It is difficult to understand what patient had to be removed. Additionally, it is unclear that four subjects had to be removed. Please describe in detail.

2.As mentioned by the authors, a p-value <0.05 was deemed statistically significant. Based on the sample size and exploratory nature of the study, we did not correct for multiple comparisons. However, you can use and analyze post hoc test with ANOVA. Please reconsider statistical analysis.

3.Figure 1 shows self-reported symptoms. The concussion symptoms and PPST in acute PTH were lower scores than the healthy control, but there were no significant differences between persistent PTH and acute PTH. Please describe the reason.

Author Response

The authors analyzed the evidence of chronic complement activation in 30 asymptomatic pediatric brain injuries. 

It is a well-planned study and I think it will contribute to the literature.

However, there are concerns particularly for technical and statistical issues.

The authors stated "Four subjects had to be removed because of sample qual-88 ity, leaving 15 participants with A-PTH, 15 with P-PTH, and 14 Healthy Controls." It is difficult to understand what patient had to be removed. Additionally, it is unclear that four subjects had to be removed. Please describe in detail.

We have provided more information that these subjects were removed because their samples did not meet minimum requirements for quantification.

As mentioned by the authors, a p-value <0.05 was deemed statistically significant. Based on the sample size and exploratory nature of the study, we did not correct for multiple comparisons. However, you can use and analyze post hoc test with ANOVA. Please reconsider statistical analysis.

Our interest in the applied approach was to provide a targeted analysis of the main groups of interest. Use of an ANOVA (omnibus test) would have decreased our sensitivity for the main group contrasts of interest (healthy to A-PTH and healthy to P-PTH). We have subsequent work being completed with larger cohort sizes where we are building in omnibus tests to address the three groups of interest; however, for this investigation we wanted to optimize our statistical sensitivity for our group comparisons of interest.

Figure 1 shows self-reported symptoms. The concussion symptoms and PPST in acute PTH were lower scores than the healthy control, but there were no significant differences between persistent PTH and acute PTH. Please describe the reason.

Thank you. There was no statistical test performed on this comparison. Our fundamental interest here was understanding how these two populations were different relative to healthy controls. Based on our lower sample size, we elected to provided very targeted statistical tests for our main analyses. Regarding the symptom expression, our interests were to understand if they were different from a healthy population. As such, there was no specific test performed between the two PTH cohorts for this analysis. Importantly, our conclusions are tied to our analyses and so we do not comment on the differential expression between cohorts, but rather relative to the healthy sample.